# Modulation of Titin and Contraction-Regulating Proteins in a Rat Model of Heart Failure with Preserved Ejection Fraction: Limb vs. Diaphragmatic Muscle

**DOI:** 10.3390/ijms25126618

**Published:** 2024-06-16

**Authors:** Beatrice Vahle, Leonard Heilmann, Antje Schauer, Antje Augstein, Maria-Elisa Prieto Jarabo, Peggy Barthel, Norman Mangner, Siegfried Labeit, T. Scott Bowen, Axel Linke, Volker Adams

**Affiliations:** 1Heart Center Dresden, Laboratory of Molecular and Experimental Cardiology, TU Dresden, 01307 Dresden, Germany; beatrice.vahle@tu-dresden.de (B.V.); leonard.heilmann@mailbox.tu-dresden.de (L.H.); antje.schauer@tu-dresden.de (A.S.); antje.augstein@tu-dresden.de (A.A.); maria-elisa.prietojarabo@herzzentrum-dresden.com (M.-E.P.J.); peggy.barthel@tu-dresden.de (P.B.); norman.mangner@tu-dresden.de (N.M.); axel.linke@tu-dresden.de (A.L.); 2DZHK Partner Site Mannheim-Heidelberg, Medical Faculty Mannheim, University of Heidelberg, 68169 Mannheim, Germany; labeit@medma.de; 3Myomedix GmbH, 69151 Neckargemünd, Germany; 4School of Biomedical Sciences, Faculty of Biological Sciences, University of Leeds, Leeds LS2 9JT, UK; t.s.bowen@leeds.ac.uk

**Keywords:** HFpEF, ZSF1, skeletal muscle dysfunction, titin, contractile proteins, diaphragm

## Abstract

Heart failure with preserved ejection fraction (HFpEF) is characterized by biomechanically dysfunctional cardiomyocytes. Underlying cellular changes include perturbed myocardial titin expression and titin hypophosphorylation leading to titin filament stiffening. Beside these well-studied alterations at the cardiomyocyte level, exercise intolerance is another hallmark of HFpEF caused by molecular alterations in skeletal muscle (SKM). Currently, there is a lack of data regarding titin modulation in the SKM of HFpEF. Therefore, the aim of the present study was to analyze molecular alterations in limb SKM (tibialis anterior (TA)) and in the diaphragm (Dia), as a more central SKM, with a focus on titin, titin phosphorylation, and contraction-regulating proteins. This study was performed with muscle tissue, obtained from 32-week old female ZSF-1 rats, an established a HFpEF rat model. Our results showed a hyperphosphorylation of titin in limb SKM, based on enhanced phosphorylation at the PEVK region, which is known to lead to titin filament stiffening. This hyperphosphorylation could be reversed by high-intensity interval training (HIIT). Additionally, a negative correlation occurring between the phosphorylation state of titin and the muscle force in the limb SKM was evident. For the Dia, no alterations in the phosphorylation state of titin could be detected. Supported by data of previous studies, this suggests an exercise effect of the Dia in HFpEF. Regarding the expression of contraction regulating proteins, significant differences between Dia and limb SKM could be detected, supporting muscle atrophy and dysfunction in limb SKM, but not in the Dia. Altogether, these data suggest a correlation between titin stiffening and the appearance of exercise intolerance in HFpEF, as well as a differential regulation between different SKM groups.

## 1. Introduction

Heart failure with preserved ejection fraction (HFpEF) counts for approximately 50% of all heart failure (HF) cases with a rising prevalence due to an increase in life expectancy and an increasing prevalence of comorbidities like obesity and diabetes [1]. Besides alterations in the myocardium like left ventricular stiffening and increased fibrosis, HFpEF is also associated with alterations in limb skeletal muscle (SKM). Alterations observed in animal models and HFpEF patients include decreased muscle mass (muscle atrophy) [2,3], changes in fiber type composition, a reduced fiber to capillary ratio [4,5,6], increased fat infiltration [6,7], and reduced mitochondrial function and content [8,9,10]. Currently, it is unclear how these changes mechanistically contribute to reduced exercise tolerance, a hallmark of all HF patients. In addition to changes occurring in limb SKM, alterations in the diaphragmatic muscle (Dia) have also been described in HF [11,12]. With respect to the situation in HFpEF, the results obtained in experimental models seem to depend on the model investigated. Analyzing a pure hypertension model (Dahl salt-sensitive rat), reduced diaphragmatic specific muscle force and increased fatigability could be detected [13], whereas in a metabolically driven HFpEF model (ZSF1 obese rat), no change in maximal force or peak power of diaphragmatic muscle strips were observed [4].

Titin, which is the largest known protein in nature, spans half of the sarcomere from the Z-disc to the M-band and is regarded as a regulator of muscle development, function, and signaling [14,15]. In particular, phosphorylation is known to have an impact on titin function and stiffness. In HFpEF, titin hypophosphorylation has been detected in the myocardium of animal models [2,16] and humans [17,18], contributing to myocardial stiffness and diastolic dysfunction. Molecular remodeling, characterized by an isoform switch from N2BA to the shorter and stiffer N2B form, together with hypophosphorylation at the N2B region and hyperphosphorylation in the PEVK region, appears to be the basis for stress-induced stiffening of titin [19,20,21].

These previous studies focused on changes at the N2B element of cardiac titin. However, in the SKM, only N2A (and not the N2B element) is present. With respect to SKM, a modulation of titin in atrophic situations like muscle unloading [22,23], chronic alcohol consumption [24,25], and space-flight [26] has been reported. In conditions of SKM unloading, for either 7 or 14 days, a significant reduction in full-length titin content was observed. Treatment with either L-arginine or exercise normalized titin content [22,23]. Also for the Dia, titin alterations are reported in atrophic conditions, like mechanical ventilation of critically ill patients [27]. Besides titin content and secondary modifications of titin through phosphorylation, acetylation, and ubiquitination, its correct anchoring to the Z-disc is crucial for proper muscle function [28]. Proteins found in the Z-disc are important for maintaining the sarcomere structure and enable the sarcomere to facilitate force transduction and signaling (for review see [29]). The importance of Z-disc proteins like α-actinin, telethonin, or myotilin is further substantiated by the observation that a loss is associated with the development of myopathies [30,31,32,33]. 

Given that HFpEF is associated with muscle atrophy and dysfunction and titin, as well as Z-disc-associated proteins, which are relevant for proper muscle function, we aimed to determine the protein expression of titin as well as its phosphorylation state in muscle samples obtained from control and HFpEF rats. Additionally, we assessed the expression of several proteins that are associated with the contractile apparatus. A schematic drawing, showing sarcomere organization and skeletal titin structure, summarizes the localization of the investigated proteins in this study (see Appendix A). Overall, we focused on a comparison between limb SKM and Dia, since, as described above, contrary regulations were observed in earlier studies. Our results suggest that, in HFpEF, alterations in the sarcomere, with impacts on titin filaments and the contractile system, also occur in SKM, possibly contributing to exercise dysfunction in HFpEF. 

## 2. Results

### 2.1. Animal Characteristics

Our group has recently reported a detailed myocardial characterization of ZSF-1 rats investigated in the present study [34]. These data showed that the obese HFpEF rats, at 32 weeks of age, had significantly higher body weight and significantly impaired diastolic function (reduction in E/é) with a preserved, left ventricular, ejection fraction, when compared to lean controls (con). Altogether, these results clearly show that the animals developed HFpEF. Regarding the SKM, the HFpEF rats showed significantly reduced muscle mass (tibialis anterior (TA) muscle weight relative to tibia length; con: 14.93 ± 0.16 mg/mm vs. HFpEF: 13.83 ± 0.16 mg/mm, *p* < 0.001). Lower muscle mass was accompanied by a lower fiber cross sectional area (CSA) (Figure 1A,B). Furthermore, the HFpEF animals exhibited reduced muscle force development of the soleus muscle (Sol), as recently reported [35]. In contrast, in the Dia, we found muscle hypertrophy with a significant increase in CSA (Figure 1C,D).

### 2.2. Expression and Secondary Modifications of Titin

To analyze titin expression, as well as secondary modifications, vertical agarose gel electrophoresis (VAGE) was performed on tissue samples of TA and Dia. As shown in Figure 2A a trend towards a lower expression of titin was evident in the TA muscle in HFpEF, whereas a significantly higher expression was noted in the Dia (Figure 2B). Lowered expression of titin in the TA was accompanied by a higher ratio of phosphorylated titin (Figure 2C). This increase in phosphorylation was not observed in the Dia (Figure 2D). To investigate titin ubiquitination, we performed a Western blot (WB) analysis, using a pan-ubiquitin antibody. The amount of ubiquitinated titin was normalized to the total amount of titin (WB analysis using N2A specific antibodies). As depicted in Figure 2E, no change in the ubiquitinated titin/titin ratio was detected in the TA, whereas a significantly higher ratio of ubiquitinated titin/titin was observed in the Dia (Figure 2F). For titin acetylation, no signal could be detected in the TA nor in the Dia.

Furthermore, we were able to detect a significant negative correlation between SKM force, as determined in an earlier study [35], and the ratio of phospho-titin/titin, for specific force (r = −0.6987, *p* = 0.0013) as well as absolute force (r = −0.5921, *p* = 0.0096).

To validate our titin detection system, we evaluated the titin phosphorylation in the Myo which, as reported in the current literature [36,37], showed a hypophosphorylation in HFpEF animals (con: 1.0 ± 0.05 vs. HFpEF: 0.82 ± 0.04, *p* = 0.003).

### 2.3. Expression and Secondary Modifications of Myosin Heavy Chain (MHC) and Nebulin

Quantifying the expression levels of myosin heavy chain (MHC) in the TA (Figure 3A) and the Dia (Figure 3B), no difference between con and HFpEF was evident. With respect to secondary modifications by phosphorylation or ubiquitination, in the TA, a significant decrease in phosphorylation (Figure 3C) and a significant increase in ubiquitination of K48 (Figure 3E) in HFpEF were detected. In the Dia, the secondary modifications of MHC were not significantly different between the two groups, but a tendency for a higher phosphorylation of MHC could be observed (Figure 3D,F). Assessment of nebulin expression, an actin-binding protein that is localized to the thin filament and the Z-disc of the sarcomeres in SKM, [38], revealed a lower expression in the TA of HFpEF animals (Figure 3G), whereas the Dia showed no differences between the groups (Figure 3H).

### 2.4. Expression of Proteins Involved in Sarcomere Organization

The expression of SMYD2, a protein playing an important function in sarcomere organization and maintenance [39] and that is essential for normal skeletal muscle function [40], is differently regulated in the TA and the Dia of HFpEF animals. In the TA, a significant downregulation of SMYD2 was detected in HFpEF animals (Figure 4A), whereas in the Dia an up-regulation was observed (Figure 4B). Regarding the expression of α-actinin, a linker between titin and the anti-parallel actin filaments, which builds a lattice-like structure that leads to a stabilization of the contractile apparatus [41], no difference was detected between con and HFpEF in the TA and the Dia (Figure 4C,D). Another important protein for structural integrity of sarcomeres at the Z-disc is Myotilin, a binding partner of α-actinin [39]. Also, for Myotilin, no difference could be detected between con and HFpEF animals, neither in the TA nor in the Dia (Figure 4E,F).

Telethonin (T-cap) anchors titin in the Z-disc through its N-terminus. T-cap is important for sarcomere development and stability [42,43]. The protein expression levels of telethonin were not different between the control and HFpEF in TA and Dia (Figure 4G,H). 

### 2.5. Expression of Ca^2+^-Related Contractile Proteins

The three subunits of troponin (Tns) (troponin C (TnC), -T (TnT), and -I (TnI)), that build a complex together with tropomyosin (Tm), are important for the Ca^2+^-dependent muscle contraction [44]. TnI, TnT, and Tm showed no differences between HFpEF and control animals, regarding their protein expression, in neither the TA muscle nor in the Dia (Figure 5C–H). However, TnC, which is responsible for calcium-binding [45], showed a strong upregulation of its expression in the Dia (Figure 5B) but not in the TA (Figure 5A).

### 2.6. Proteins Related to Muscle Atrophy

Since HFpEF causes chronic stress on muscles, we next looked for atrophy-related proteins. We investigated the expression of muscle RING-finger protein-1 (MuRF-1), Muscle Atrophy F-box (MAFbx), four and a half LIM domain 1 (FHL1), and Myostatin (GDF8). For MuRF-1, an E3-ubiquitin ligase that interacts with titin and nebulin [46], a significant upregulation of protein expression could be found in the TA of HFpEF animals. In the Dia, no difference could be detected between control and HFpEF. On the other hand, MAFbx, an E3-ubiquitin ligase that acts in a similar way to MuRF-1 [46], showed no change in its expression, neither in the TA nor in the Dia (Figure 6A–D). 

The third atrophy-related protein, FHL1, is a scaffolding protein that is supposed to play a role in autophagy-processes in the skeletal muscle [47,48]. In the Dia, no difference in the protein expression level between the control and HFpEF was evident; a significant upregulation could be detected in the TA (Figure 6E,F).

The last atrophy-marker we investigated was GDF8. GDF8 belongs to the growth factor-β superfamily and exhibits transforming characteristics. In the TA, we were able to detect a significant upregulation in GDF8 expression, whereas a significant downregulation was evident in the Dia (Figure 6G,H), which aligns with the atrophy and hypertrophy observed in these respective muscles.

### 2.7. Site-Specific Phosphorylation of Titin and Impact of Exercise Training on Titin Phosphorylation

To gain further insights into the mechanistic regulation of SKM titin in HFpEF, we investigated titin phosphorylation at the PEVK-site (S11878), since increased phosphorylation in this region is known to be associated with fiber stiffness [49,50]. As shown in Figure 7A, we were able to detect significantly higher S11878 phosphorylation in HFpEF when compared to con.

To clarify, if the observed titin hyperphosphorylation in the limb SKM was, at least partly, elicited by muscle disuse, we used HFpEF animals from an earlier training study [51,52]. The aim was to investigate if 8 weeks of high intensity interval training (HIIT) was able to reverse hyperphosphorylation. As shown in Figure 7B, the hyperphosphorylation occurring in HFpEF could be reversed by HIIT.

## 3. Discussion

A proven hallmark of HFpEF is exercise intolerance, accompanied with functional, as well as molecular alterations in limb SKM. This disease is also associated with muscle dysfunction, early fatigue and muscle atrophy, which can be partly reversed by exercise training [53,54]. In contrast, the Dia, as a respiratory SKM, shows no alterations regarding skeletal muscle function in an HFpEF model [4,48,55]. Titin is an important protein for the regulation of muscle development and contraction. For the myocardium, it is well described that HFpEF models show hypophosphorylation of titin, accompanied by increased stiffness [17,56]. However, to date, little is known about titin modifications in the SKM in HFpEF [57]. In the present work, we evaluated the impact of HFpEF on SKM titin and contraction-regulated proteins, assessing secondary modifications and protein expression. We compared the molecular characteristics of limb and respiratory SKM, obtained from a validated animal model of HFpEF. The results of the present study can be summarized as follows:In the limb SKM, we observed a hyperphosphorylation of titin, accompanied by reduced titin expression. Specifically the phosphorylation of S11878 in the PEVK region of titin was elevated in HFpEF.Titin content and phosphorylation, secondary modifications of MHC, and the expression of nebulin and proteins involved in sarcomere organization (SMYD2) and muscle atrophy (MuRF1, FHL1, and GDF8) are differently regulated between Dia and limb SKM in HFpEF.Exercise training via HIIT was able to reverse titin hyperphosphorylation in limb SKM in HFpEF.

Taken together, these results suggest that in HFpEF titin phosphorylation in skeletal muscle differs significantly when compared to the myocardium, where titin hypophosphorylation is reported [36,37]. In addition, we can conclude that these changes seem to be partly inactivity-induced, since the given titin modifications were reversible following prolonged exercise training.

### 3.1. Titin Phosphorylation and Skeletal Muscle Function

Titin is described, not only as a passive but also an active force generator, playing an important role in contractile mechanisms [14,58]. Molecular mechanisms like phosphorylation, Ca^2+^ binding, and actin-titin interaction, as well as chaperone binding and oxidation, are known to regulate the mechanical properties of titin [58]. There are four main kinases, which phosphorylate titin: PKA, PKG, PKC, and CamKIIδ. PKA and PKG mainly phosphorylate the N2B-region, leading to a more elastic titin, while PKC and CamKIIδ are known to stiffen titin through phosphorylation at the PEVK-region [19].

In the myocardium, titin stiffness is regulated through an isoform switch from N2BA to the shorter and stiffer N2B form, along with hypophosphorylation. This hypophosphorylation occurs mainly at the N2B region, while the titin PEVK region, on the other hand, shows hyperphosphorylation at the S11878 site [19,59,60].

In our study, we showed hyperphosphorylation of limb skeletal muscle titin for the TA and Qua, along with slightly reduced titin content, in an HFpEF animal model. We were able to assign this phosphorylation shift to hyperphosphorylation in the PEVK-region at the S11878 phosphorylation site, which is known to increase passive stiffness [61,62]. This finding aligns with studies that reported titin-based stiffening of muscle fibers, in multiple disease, caused by hyperphosphorylation of S11878 [63]. Since there is no N2B region in the SKM, the second main phosphorylation site of titin is probably the N2A region. Not much is known about phosphorylation sites in N2A. Recently, it was reported that PKA mediates N2A phosphorylation but causes, in contrast to N2B phosphorylation, only a small reduction in passive force [64]. This means that phosphorylation at the PEVK-region has strong negative effects on titin elasticity, N2B phosphorylation has some positive effects regarding titin elasticity, and N2A phosphorylation has nearly no beneficial effects on titin function. 

In addition, we were able to identify a negative correlation between the phosphorylation state of titin and muscle force. This means that increased titin phosphorylation was associated with less force production, such that a modulation in titin phosphorylation might influence exercise performance. This assumption is supported by the intervention of exercise training in our HFpEF model, as this normalized phosphorylated titin alongside the ~15% increase in VO_2_peak, indicating better cardiorespiratory fitness, which was previously reported in these animals [51]. Furthermore, this effect of exercise training on titin phosphorylation also indicates that the hyperphosphorylation of titin in HFpEF might be triggered by disuse.

In limb SKM, a clear hyperphosphorylation of titin was evident; however, we were not able to see any change in the phosphorylation state of titin in the Dia. In addition, we detected an upregulation in the total titin content. Others have similarly found an upregulation of titin in Dia, when introducing exercise to their HFpEF mouse model [57]. Altogether, this supports the hypothesis of a ‘training effect’, due to increased breathing, in the ZSF-1 animal model [4,13]. 

This leads us to the assertion that, in contrast to the Myo, hyperphosphorylation in SKM is linked to increased titin stiffness, which seems to be mainly disuse-induced and can be reversed by exercise training. This hyperphosphorylation primary takes part in the PEVK-region of titin. Since, in the SKM, titin cannot adapt its elasticity through N2B phosphorylation, it seems as if the regulation of stiffness and elasticity is mainly caused by (de-)phosphorylation of the PEVK-region.

### 3.2. Contraction-Regulating Proteins—Regulation in HFpEF

Titin is supposed to have a big interactome, divided into proteins interacting with the Z-disc, I-band, M-band, and A-band [65]. Titin-based molecular and conformational changes can lead to different alterations, regulating protein synthesis and degradation [14]. Therefore, we investigated proteins known to interact with titin and observed the expression changes in our HFpEF-animal model.

MHC is a direct interaction partner of titin’s A-band [65]. For total MHC, no changes in protein expression were evident, but significant alterations in phosphorylation and ubiquitination as secondary modifications were observed. Especially in limb SKM, reduced phosphorylation and enhanced ubiquitination were observed. In contrast, the Dia showed no changes regarding ubiquitination and a tendency towards a higher phosphorylation. The relevance of these secondary modifications, regarding the structural and enzymatic function of MHC remains speculative. In a recently published manuscript, Landim-Vieira and colleagues reported that MHC hypophosphorylation at S210 and T215 alters ADP-release, ATPase activity, and sliding velocity [66]. In addition, hypophosphorylation is also associated with perturbed cross-bridge kinetics [67]. Unfortunately, these experiments were performed in myocardial tissue, so no data are available for SKM. 

Regarding the preservation of muscle function, nebulin, an 800 kDa protein [68], is crucial. It interacts with the Z-disc of titin, is incorporated with the thin filament of the sarcomere, and is supposed to be inextensible. Moreover, little information is available for nebulin, since many mutations make it hard to predict functional roles as well as genotypic and phenotypic outcomes [38]. In our HFpEF rat model, we observed a downregulation of nebulin in the TA, which may contribute to impaired muscle function.

SMYD2 is a lysine methyltransferase [69], implicated to be important for titin stability and normal skeletal muscle function [40]. In HFpEF, a significant downregulation in the TA but an upregulation in the Dia could be observed. This follows the suggestion that a reduction in SMYD2 expression has a structurally negative effect on the N2A domain of titin [40], consistent with our results of a titin reduction in the TA and an increase in the Dia. 

Regarding muscle contraction and Ca^2+^-sensing, the troponin–tropomyosin complex, which interacts with multiple regions of titin, is crucial for normal function. Troponin is a hetero-trimer, consisting of three Isoforms: TnC, TnI, and TnT [70,71,72]. TnC is the Ca^2+^ binding part of this complex, TnI plays a role as an inhibitory unit and TnT is important for binding tropomyosin, which interacts with titin. All three troponin subunits have different fiber-type-specific isoforms, which are expressed according to the muscle type. Especially for TnC, a close correlation between MHC- and TnC-isoform expression could be detected [70,73]. Only TnC showed an upregulation in the Dia, while the other proteins remained unaltered. In the literature, TnC is mostly described as a calcium-sensing switch [73]. Depending on the occurring fiber type, two different isoforms of TnC are expressed. While fast TnC possesses two calcium-binding sites, the slow isoform has only one binding-site [73]. Although TnC is not described as the limiting factor in this complex [70], still we see an upregulation in the Dia. We can only speculate about the mechanistic background behind this TnC upregulation, but a possible explanation could be a calcium overload, due to increased work and a potential leak reported in other forms of heart failure [12]. Another supporting argument for this hypothesis could be the fiber type switch, towards a slow fiber type, that is evident in the Dia in HFpEF [4]. It has been suggested, that a TnC-isoform switch towards slow-TnC occurs, along with a reduction in calcium binding sites [73].

In addition to impaired muscle force and function, our HFpEF model showed muscle atrophy in the TA, along with reduced CSA. To investigate the molecular mechanism, we examined the expression of atrophy-related proteins, which interact with titin. MuRF1 and MAFbx are atrophy-related E3-ubiquitin-ligases [74,75], whereas FHL1 is a scaffolding protein able to modulate myostatin expression [47].

As expected, MuRF1 was significantly upregulated in the TA of HFpEF rats, but, surprisingly, MAFbx expression was not altered in HFpEF. However, both E3-ligases are known interaction partners of titin that may influence titin content through degradation via the autophagosome [75]. MuRF1, in particular, is known to stabilize titin and, in case of an overexpression, it disrupts titin’s C-terminal domain [76]. This correlates with the slightly reduced titin expression, as observed in the present study. 

FHL1 is an atrophy-related protein that, due to mutations, is associated with severe muscular dystrophies. Upregulation in FHL1 expression in the myocardium is normally associated with hypertrophy and reduced FHL1 expression with atrophy [77]. On the other hand, FHL1 is supposed to activate myostatin in the SKM, resulting in the activation of autophagy-associated genes and autophagy, along with increased MuRF1 and MAFbx expression [47]. 

Consistent with increased FHL1 expression in the TA, we were also able to detect an upregulation of myostatin in the TA. As described, myostatin, which belongs to the growth factor-β superfamily, becomes activated by FHL1 and is known to negatively regulate SKM growth [78]. Furthermore, it is associated with autophagy activation as well as MuRF1 and MAFbx expression [47]. For the Dia we detected, according to the observed hypertrophy and the hypothesis of an exercise effect in HFpEF, a significant upregulation of myostatin was observed in the HFpEF animal.

Other structural Z-discs proteins we investigated, like α-actinin, Myotilin, and telethonin, remained unaltered in HFpEF. A summary of all protein changes, in response to HFpEF, within the current study are provided in Table 1. 

### 3.3. Limb SKM vs. Diaphragm

In our study, we not only detected differential regulation of titin in HFpEF, between the heart muscle (Myo) and SKM, but also a divergent response between limb SKM (TA) and Dia.

While the TA showed strong signs of muscular dysfunction, along with reduced sarcomere stability and atrophy, the Dia showed signs of improved muscle function, like hypertrophy and evidence for increased sarcomere stability. These results were accompanied by changes in titin expression.

In the TA, titin hyperphosphorylation, together with a reduced titin amount, was evident in HFpEF, while the Dia showed no differences regarding titin phosphorylation but a strong upregulation of titin expression in HFpEF.

For proteins playing an important role in titin function and stabilization, like SMYD2 and nebulin, the results for Dia and limb SKM were opposite. While the Dia showed signs of improved titin stability, we were able to find evidence for decreasing stability of titin in limb SKM. These findings correlate with the overall titin levels in the muscle groups.

Furthermore, the limb SKM exhibited muscle atrophy in HFpEF, whereas muscle hypertrophy was evident in the Dia. This aligns with the modulation of atrophy-related proteins in the TA but not, or even in the opposite way, in the Dia.

Altogether, these findings indicate that the limb SKM shows strong negative effects, regarding sarcomere function in HFpEF, while the Dia appears to be protected against these detrimental changes [4]. Regarding earlier research, this seems to be caused by the increased breathing frequency in HFpEF [79], causing an exercise training effect in the Dia. In contrast, the limb SKM shows strong signs for muscle wasting, at least partly disuse-induced, resulting in restricted muscle function and sarcomere stability. 

### 3.4. Study Limitations

Despite our innovative findings regarding skeletal muscle alterations, with a focus on titin in HFpEF, some limitations have to be mentioned.

First, for limb muscle characterizations we used different muscle groups from the leg (TA, Qua, and Sol). However, many studies showed an overall dysregulation in the limb muscle, independent of specific muscle groups [35,80].

Second, we only investigated changes in ZSF1 rats, which is an artificial model of HFpEF, triggered by metabolic alterations. However, we did not cover the whole spectrum of factors that are associated with the expression of the disease, as performed in clinical practice. Therefore, a generalization to other forms of HFpEF (e.g., hypertension triggered HFpEF (Dahl salt-sensitive rats)) is not given. However, in an earlier study, we investigated several animal models of HFpEF [81] and concluded that the ZSF1 model is the most suitable for studying SKM alterations in HFpEF.

## 4. Materials and Methods

### 4.1. Animals

Female ZSF1 lean (n = 17) (control) and obese (n = 17) (HFpEF) rats (Charles River, Sulzfeld, Germany) were included in the present study. At the age of 32 weeks, the animals were sacrificed, muscle function (Sol) was assessed, and muscle tissue (TA, Dia and Myo) was snap frozen and stored at −80 °C or fixed with 4% PBS buffered formalin and embedded in paraffin. A detailed description of the study design and animal characteristics is given in a recently published study by our group [34]. Additionally, tissue from the Qua, obtained from a training study with male ZSF1 rats published by our group [82], was used.

### 4.2. Analysis of Titin and Nebulin

To analyze the expression of titin and nebulin protein, homogenates of TA, Dia, Myo, and Qua were resolved on 1% agarose gels, as originally described by [31]. In brief, pulverized tissue was solved in urea buffer (8 mol/L urea, 2 mol/L thiourea, 0.05 mol/L Tris pH 6.8, 0.075 mol/L DTT, and 3% SDS) including a protease and phosphatase inhibitor mix (Serva, Heidelberg, Germany) at a ratio between 1:40 and 1:60 (weight/volume), carefully inverted several times and heated for 10 min at 60 °C. After this, glycerol (final concentration 25%) was added, followed by a centrifugation step (10 min at 13.200× *g*). The supernatant was collected, traces of bromophenol blue were added, and the samples were stored at −80 °C. Protein concentration was measured by using the 660 nm protein assay (Thermo Fischer Scientific, Waltham, MA, USA). For electrophoresis 3 (TA), up to 6 µg (Dia) of protein was loaded onto the gel and run for 5 h at 15 mA with cooling. For the assessment of phosphorylated titin and MHC, the gels were stained with Pro-Q™ Diamond Phosphoprotein Gel Stain, followed by Sypro Ruby gel stain for total titin expression (both Thermo Fischer Scientific, Waltham, MA, USA), according to the manufacturer’s recommendation. For the assessment of protein expression, densitometry was performed using the 1D scan software package version 15.08b (Scanalytics Inc., Rockville, MD, USA). Measurements for total titin and nebulin were normalized to MHC expression.

To assess nebulin and titin modifications by acetylation and ubiquitination, proteins were transferred to a polyvinylidene fluoride membrane (PVDF) using a tank blot (140 min at 40 V). The following antibodies were used: poly-ubiquitin (Cell Systems, Leiden, The Netherlands; #58395; 1:1000); acetylated-lysine (Cell Signaling, 9441S, 1:1000); titin N2A (Myomedix, Neckargemünd, Germany, 1 µg/mL); and pSer11878 (generously provided by Prof. Dr. W. Linke, University Münster, 1:1000) and the ratio to unmodified protein was calculated. All data are presented as x-fold change, relative to control. Densitometry was used for quantification (1D scan software package version 15.08b; Scanalytics Inc., Rockville, MD, USA).

### 4.3. Histological Analyses

For the assessment of CSA, paraffin-embedded TA and Dia were sectioned (4 µm), mounted on glass slides and stained with hematoxylin and eosin. The fiber CSA was evaluated by imaging software (Zen imaging software, Zeiss, Jena, Germany). A minimum of 200 fibers per section were measured.

### 4.4. Western Blot Analyses

For protein quantification Western blot analyses were performed. Frozen muscle tissue was homogenized in RIPA buffer (50 mmol/L Tris pH 7.4, 1% NP-40, 0.25% Na-deoxycholate, 150 mmol/L NaCl, and 1 mmol/L EDTA) including a protease inhibitor mix (Inhibitor mix M, Serva, Heidelberg, Germany). Protein concentration was determined (BCA assay, Pierce, Bonn, Germany) and aliquots (10 μg) were separated by SDS-polyacrylamide gel electrophoresis. Proteins were transferred to a PVDF membrane and incubated overnight at 4 °C, using the following primary antibodies: SMYD2 (21290-1-AP, 1:2000), α-actinin (14221-1-AP, 1:2000), MYOT (10731-1-AP, 1:2000), FHL1 (10991-1-AP,1:1000), GDF-8 (19142-1-AP, 1:1000), tropomyosin (11038-1-AP, 1:2000), troponin C (13504-1-AP, 1:1000), troponin T (15513-1-AP, 1:1000) (all Proteintech, Planegg-Martinsried, Germany), troponin I (ab184554, 1:2000), telethonin (ab133646, 1:1000), MAFbx (ab168372, 1:1000) (all Abcam, Cambridge, UK), and MuRF-1 (Santa Cruz, Dallas, TX, USA, sc398608, 1:200). Membranes were subsequently incubated with a horseradish peroxidase-conjugated secondary antibody and specific bands were visualized by enzymatic chemiluminescence (Super Signal West Pico, Thermo Fisher Scientific Inc., Bonn, Germany). Densitometry was used for quantification (1D scan software package version 15.08b; Scanalytics Inc., Rockville, MD, USA). Measurements were normalized to the loading control GAPDH (1:10,000; HyTest Ltd., Turku, Finland) or to overall protein loading, as determined by Ponceau S staining. All data are presented as x-fold change, relative to the control.

### 4.5. Statistical Analyses

Statistical analysis was performed using a *t* test, a Kruskal–Wallis Test or an ANOVA followed Tukey post-test analysis, as appropriate. All findings are reported as means ± standard error of the mean (SEM). Significance was accepted as *p* < 0.05.

## 5. Conclusions

In conclusion, the results of the present work show that there is a significant difference in titin regulation between myocardial and SKM alterations in an HFpEF-model. In contrast to the myocardium, we observed a hyperphosphorylation of titin in the limb SKM, as well as reduced titin levels. This hyperphosphorylation primarily affected the PEVK region of titin. This follows signs of reduced titin stability and an upregulation of atrophy-related proteins. It seems that this effect might be related to inactivity, since exercise training reversed titin hyperphosphorylation. 

Additionally, this study strengthens the hypothesis that HFpEF exerts exercise training effects on the Dia, induced through increased breathing. This assumption was supported through contrary molecular alterations compared with limb SKM, including upregulation of titin, unaltered titin phosphorylation-levels, increased titin stability, and hypertrophy.

## Figures and Tables

**Figure 1 ijms-25-06618-f001:**
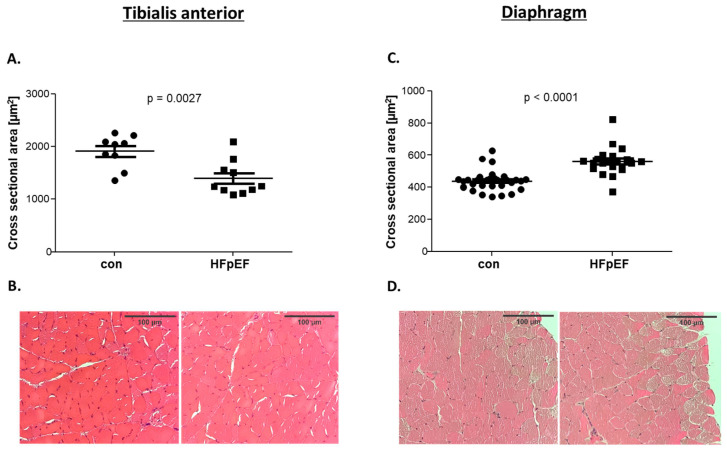
Cross sectional area (CSA) was determined in the TA (**A**) and the Dia (**C**) of control (con) and HFpEF animals (HFpEF) at the age of 32 weeks. Representative images used for the analyses of the TA (**B**) and the Dia (**D**) are depicted. The results are expressed as mean ± SEM (n = 9–30).

**Figure 2 ijms-25-06618-f002:**
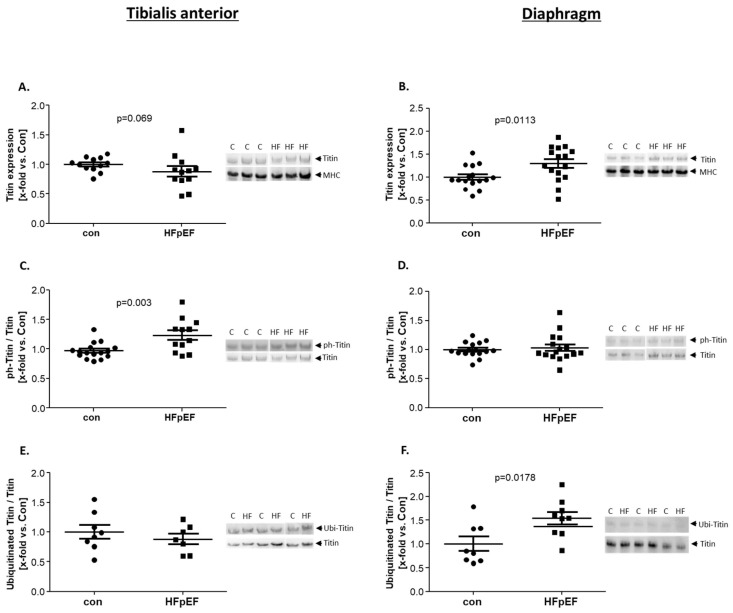
Protein expression (**A**,**B**) and secondary modifications of titin. Phosphorylation (**C**,**D**) and ubiquitination (**E**,**F**) were quantified by VAGE analysis in skeletal muscle homogenates of TA (**A**,**C**,**E**) and Dia (**B**,**D**,**F**), obtained from ZSF1-control (con, black circles) and ZSF1-HFpEF rats (HFpEF, black squares) at the age of 32 weeks. The results are expressed as x-fold vs. control, (n = 7–16 per group). Representative stains and VAGE blots are depicted (c = con, HF = HFpEF).

**Figure 3 ijms-25-06618-f003:**
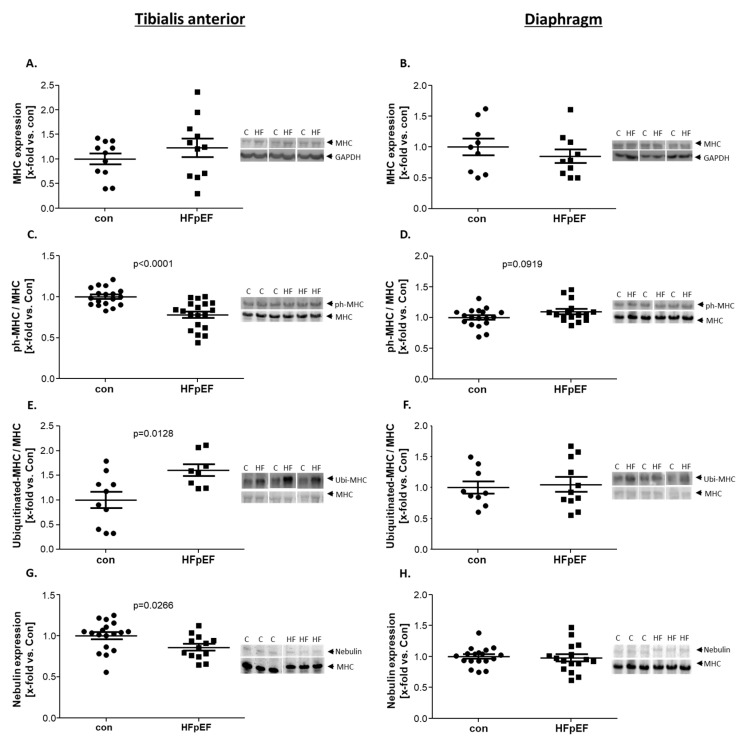
Protein expression (**A**,**B**) and secondary modifications of MHC-like phosphorylation (**C**,**D**) and ubiquitination (**E**,**F**)) and protein expression of nebulin (**G**,**H**) were quantified by WB and VAGE analysis in skeletal muscle homogenates of TA (**A**,**C**,**E**,**G**) and Dia (**B**,**D**,**F**,**H**) obtained from ZSF1-control (con, black circles) and ZSF1-HFpEF rats (HFpEF, black squares) at the age of 32 weeks. The results are expressed as x-fold vs. control (n = 9–17 per group). Representative stains and blots are depicted (c = con, HF = HFpEF).

**Figure 4 ijms-25-06618-f004:**
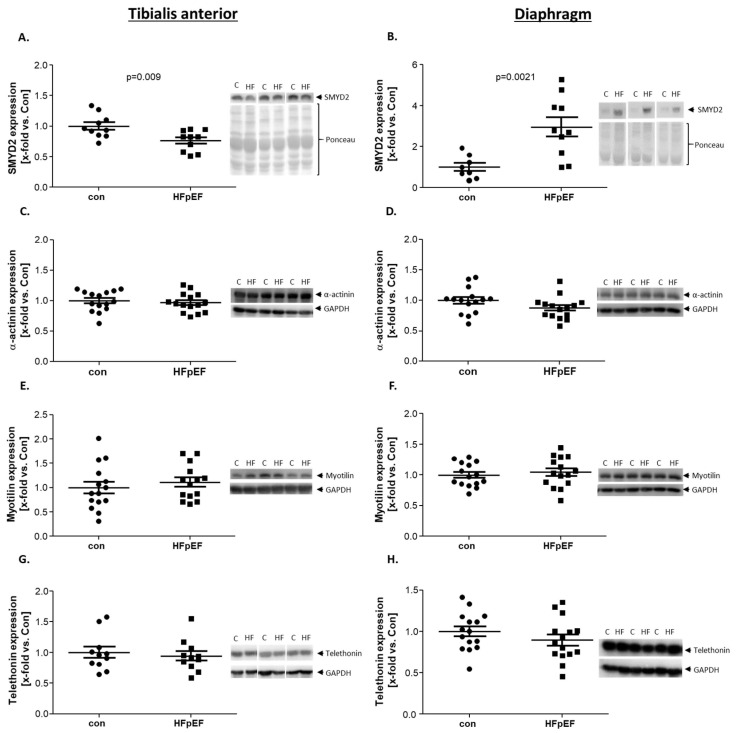
Expression of proteins involved in sarcomere organization was quantified by WB analysis of skeletal muscle homogenates of TA (**A**,**C**,**E**,**G**) and Dia (**B**,**D**,**F**,**H**) obtained from ZSF1-control (con, black circles) and ZSF1-HFpEF rats (HFpEF, black squares) at the age of 32 weeks. As contractile proteins, SMYD-2 (**A**,**B**) and α-actinin (**C**,**D**) were measured and as Z-disc proteins MYOT (**E**,**F**) and telethonin (**G**,**H**). The results are expressed as x-fold vs. control (n = 8–15 per group). Representative blots are depicted (c = con, HF = HFpEF).

**Figure 5 ijms-25-06618-f005:**
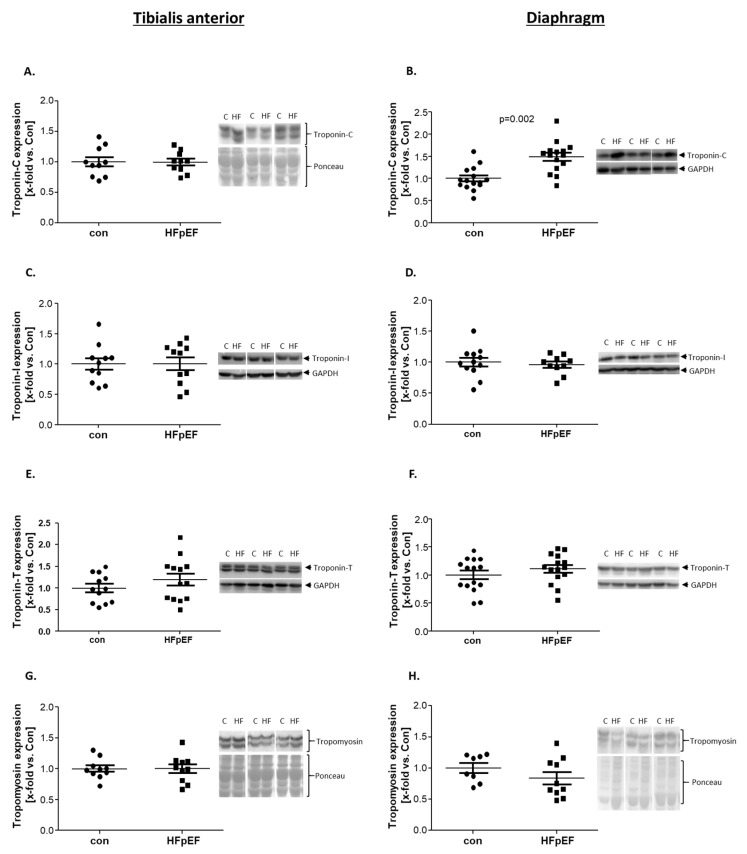
Expression of proteins related to Ca^2+^-dependent muscle contraction was quantified by WB analysis of skeletal muscle homogenates of TA (**A**,**C**,**E**,**G**) and Dia (**B**,**D**,**F**,**H**) obtained from ZSF1-control (con, black circles) and ZSF1-HFpEF rats (HFpEF, black squares) at the age of 32 weeks. As proteins, troponin-C (**A**,**B**), troponin-I (**C**,**D**) troponin-T (**E**,**F**), and tropomyosin (**G**,**H**) were measured. The results are expressed as x-fold vs. control (n = 8–15 per group). Representative blots are depicted (c = con, HF = HFpEF).

**Figure 6 ijms-25-06618-f006:**
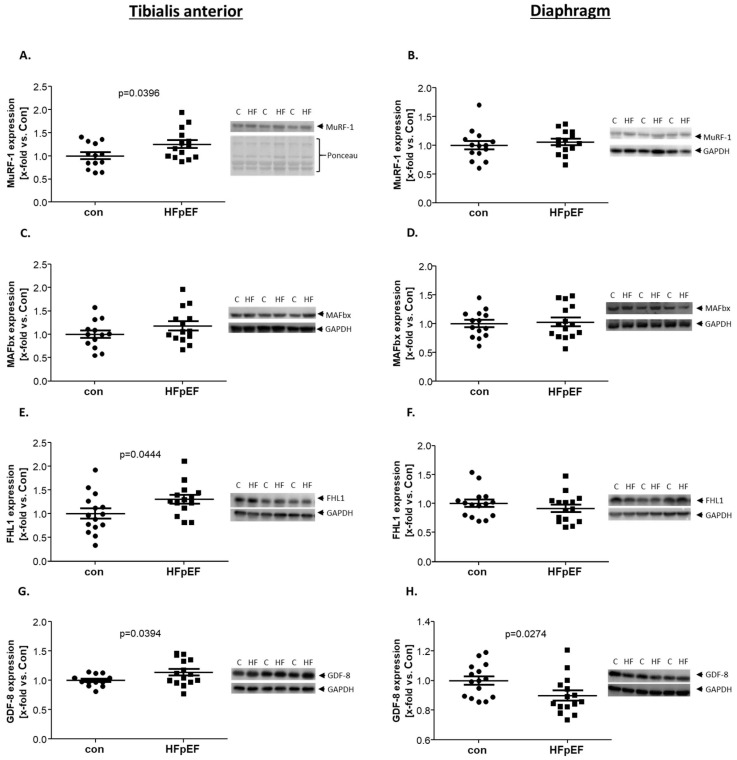
Protein expression of atrophy-related proteins was quantified by WB analysis of skeletal muscle homogenates of TA (**A**,**C**,**E**,**G**) and Dia (**B**,**D**,**F**,**H**) obtained from ZSF1-control (con, black circles) and ZSF1-HFpEF rats (HFpEF, black squares) at the age of 32 weeks. As atrophy-related proteins, MAFbx (**A**,**B**), MuRF1 (**C**,**D**), FHL1 (**E**,**F**), and GDF-8 (**G**,**H**) were measured. The results are expressed as x-fold vs. control (n = 12–15 per group). Representative blots are depicted (c = con, HF = HFpEF).

**Figure 7 ijms-25-06618-f007:**
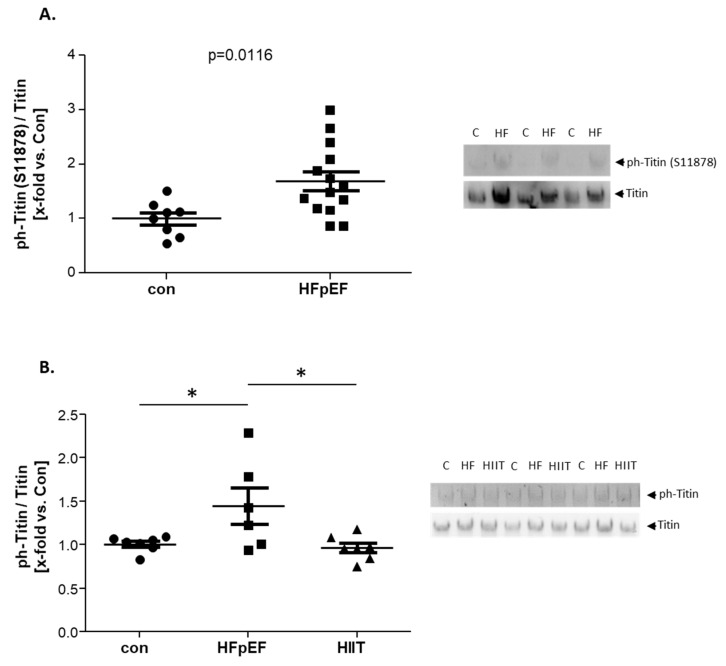
Site-specific titin phosphorylation (**A**) and the impact of exercise training (HIIT) (**B**). The PEVK phosphorylation at S11878 in TA homogenates obtained from ZSF1-control (con, black circles) and ZSF1-HFpEF rats (HFpEF, black squares) at the age of 32 weeks is depicted (**A**). Furthermore, the impact of exercise training on titin phosphorylation was analyzed in quadriceps muscle (Qua) homogenates obtained from ZSF1-control (con, black circles) and ZSF1-HFpEF rats without (HFpEF, black squares) and with HIIT (HIIT, black triangles) at the age of 28 weeks (**B**). The results are expressed as x-fold vs. control (n = 6–14 per group). Representative stains and blots are depicted (c = control, HF = HFpEF, HIIT = HFpEF + HIIT). * *p* < 0.05.

**Table 1 ijms-25-06618-t001:** Summarized characteristics and protein levels comparing TA (limb SKM) and Dia.

	TA	Dia	Figure
CSA	↓	↑	Figure 1
Titin	↘	↑	Figure 2
Phospho-Titin	↑	↔	
Ubi-Titin	↔	↔	
MHC	↔	↔	Figure 3
Phospho-MHC	↓	↗	
Ubi^K48^-MHC	↑	↔	
Nebulin	↓	↔	
SMYD2	↓	↑	Figure 4
α-actinin	↔	↔	
Myotilin	↔	↔	
Telethonin	↔	↔	
Troponin-C	↔	↑	Figure 5
Troponin-I	↔	↔	
Troponin-T	↔	↔	
Tropomyosin	↔	↔	
MuRF-1	↑	↔	Figure 6
MAFbx	↔	↔	
FHL-1	↑	↔	
GDF8	↑	↓	

↑ significant upregulation, ↓ significant downregulation, ↘ tendency towards downregulation, ↗ tendency towards upregulation, and ↔ no changes in HFpEF compared to the control.

## Data Availability

The data that support the findings of this study are available from the corresponding author upon reasonable request.

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
