# Peer review of "Modulation of Titin and Contraction-Regulating Proteins in a Rat Model of Heart Failure with Preserved Ejection Fraction: Limb vs. Diaphragmatic Muscle"

_ijms, 2024, doi:10.3390/ijms25126618_

Round 1

Reviewer 1 Report

Comments and Suggestions for Authors

This manuscript examines the expression and post-translational modification of a range of sarcomeric and sarcomere-associated proteins in skeletal muscle in response to heart failure with preserved ejection fraction (HFpEF), using an appropriate rat model. HFpEF is arguably one of the most common emerging chronic disease entities affecting western countries, so these data are highly relevant. Importantly, the authors demonstrate that atrophy occurs in limb skeletal muscle, while the diaphragm is preserved. They reasonably conclude that the changes in skeletal muscle are likely to be a consequence of a relative lack of exercise, consequent on the limited exercise tolerance in HFpEF, while the preserved function of the diaphragm may be a consequence of the increased use of the diaphragm consequent on the greater breathing exertion required in HFpEF.

The quantitative experiments have been carefully and exhaustively undertaken and represent a very substantial body of experimental data that is new, novel and relevant to understanding the pathogenesis of HFpEF. The authors recognise the limitations of their rat model and an obvious next step would be to examine and confirm changes in human diseased tissue – a difficult task due to obvious access limitations. This manuscript lays excellent groundwork for future human studies.

I have only a few minor comments that the authors may wish to address in a future version of the manuscript. While the main focus has rightly been placed on titin changes and modifications, the authors have considered a range of other proteins, including myosin heavy chain (MHC). The authors have provided relevant background for the changes in other proteins they have considered, allowing the interpretation of changes in terms of functional changes. However, I was unable to find an explanation for the consequences of the changes they observed for the MHC. Admittedly this is a difficult task, since few studies seem to exist on the consequences of altered phosphorylation of the MHC (as opposed to the MLCs, which are very well documented). For example, a recent paper considers post-translational modifications (PTMs) in cardiac b-MHC (Landim-Vieira et al. (2022) Post-translational modification patterns on β-myosin heavy chain are altered in ischemic and nonischemic human hearts eLife 11:e74919. https://doi.org/10.7554/eLife.74919). This paper makes the point that phosphorylation has the capacity to alter MHC enzymatic and structural function (in the heart) in a number of different ways, depending on the location of the phosphorylation site. They particularly focus on the 210-215 region of the head domain, but other phosphorylation sites/functions exist on MHC. In this manuscript the authors only considered the overall phosphorylation status of the MHC, using a phospho-protein stain. Localisation of the sites of hypophosphorylation that have been observed in this study could be undertaken in future studies, but I feel it would be valuable for the authors of the current manuscript to make the point that MHC hypophosphorylation is likely detrimental, consistent with their observations in limb skeletal muscle, and support this assertion with appropriate references. This could be inserted at approx. line 331 of the Discussion. The significance of ubiquitination of MHC could also be discussed.

Other minor suggested typos

1. Line 39: “…with a RISING prevalence…”

2. Line 254 and line 262: replace “going along with” with “accompanied by”

3. Line 331: “MHC is a direct partner of…”

Reviewer 2 Report

Comments and Suggestions for Authors

The Authors have carried an impressive amount of work requiring technical skills, competence in mastering biochemical assays and articulated scientific background which I respectfully recognize. However, in my opinion so much effort has not delivered much for the following reasons:

1. HFpEF is a multifactorial entity and researchers need to decide what risk factor or combination of risk factors disease should be included in their protocol design. In this perspective, the ZSF1 rat model deals primarily with the cardiometabolic phenotype of HFpEF which obviously does not cover all the spectrum of factors associated with the expression of disease as met in clinical practice.

2. At this point we still ignore whether a similar biochemical pattern would emerge or not from other animal models of HFpEF as well as HFrEF or physically deconditioned models.

3. The main message I got from this article is that the abnormalities of skeletal muscle of ZSF1 rats are secondary to a reduced level of activity since the biochemical profile of the tibial anterior muscle diverged from that of the diaphragm as a consequence of the increased level of respiratory activity imposed by the failing cardiac function. Rather unequivocal support to this conclusion comes from the effect of intensive exercise training in line with human results showing the benefits of regular exercise training in HFpEF patients and, to some albeit not directly related extent, with the recent randomized intervention trials showing the effect of weight loss, a fundamental question not addressed in this study.

The limitations as above outlined should be acknowledged more openly than how the Authors have done in their final paragraphs.
